# Towards Principled Benchmarking of Non-tabular Reinforcement learning

## Abstract

A thorough evaluation of the performance of reinforcement learning agents is critical to establish significant progress in the field, with benchmarks being the key component of this process. In the tabular setting, a rich theory of environment hardness has recently been leveraged to design benchmarks with precise characterizations of hardness. In contrast, the non-tabular setting currently lacks such a theory and instead relies on expert judgments and community popularity to establish benchmarks. This reliance on subjective assessments can limit the rigor and reliability of the evaluation process. The goal of this paper is to take the first step towards the design of principled non-tabular benchmarks by four main contributions. First, we review the theory of hardness in tabular and non-tabular settings to highlight promising directions. Second, we identify the essential features that a principled benchmarking library for non-tabular reinforcement learning should possess while explaining the limitations of existing libraries in meeting those needs. Third, we propose a new library (`pharos`) specifically designed to support the development of principled benchmarking. Finally, we present an in-depth case study that, in addition to illustrating examples of the kind of analysis that `pharos` facilitates, demonstrates that, while tabular measures can represent a component in quantifying non-tabular hardness, it is necessary to develop measures tailored for the non-tabular setting.

## 1 Introduction

Reinforcement learning (RL) is a subfield of machine learning where an agent learns to make decisions by interacting with an environment, typically with limited prior knowledge of its dynamics. Unlike supervised learning, where models are trained on labeled data, RL involves an agent exploring the environment through trial and error to discover actions that maximize cumulative reward. As the agent interacts with the environment, it receives observations representing the current state, selects actions according to its policy, and receives feedback as rewards.

Reinforcement learning agents have achieved notable success in domains with efficient simulators, e.g., board games and video games (Silver et al., 2018; Vinyals et al., 2019), where vast amounts of data can be easily collected. Nevertheless, their high sample complexity, i.e., requiring many interactions with the environment to learn an optimal policy, presents a major obstacle in more practical applications like finance, healthcare, and robotics. In these domains, each interaction with the environment incurs a significant cost, either in terms of time, resources, or potential risk. For example, in healthcare, collecting real-world data is both expensive and constrained by stringent safety regulations, while in robotics, continuous experimentation can result in equipment degradation or failure. Without efficient simulators, the high cost of acquiring sufficient samples makes traditional reinforcement learning approaches infeasible in these settings. Consequently, enhancing the sample efficiency of RL agents is essential to enable their deployment in these high-stakes, real-world applications.

Significant research in reinforcement learning has focused on reducing sample complexity, either by incorporating prior knowledge into the learning process (Wang et al., 2024) or by developing more efficient exploration strategies (Osband et al., 2019). However, a critical factor in advancing these developments is the

meticulous evaluation of agent performance, which is particularly challenging due to the stochastic nature of environments (Machado et al., 2018), the variability in agent behaviour (Jordan et al., 2020), and the complexities of defining meaningful performance metrics (Mendez et al., 2022). A rigorous evaluation process is therefore an essential prerequisite to accurately measure progress and ensure that reported improvements are both significant and generalizable.

The core element of agent evaluation lies in the design of benchmarks, which must offer a sufficiently diverse range of challenges, be large-scale enough to generate enthusiasm within the research community, and provide meaningful insights into the strengths and weaknesses of the agents being tested. In the tabular reinforcement learning setting, significant progress has been achieved by leveraging the well-established theory of hardness to create principled benchmarks (Conserva & Rauber, 2022). This ensures a more rigorous evaluation by selecting environments with diverse hardness profiles characterized by theoretical hardness measures. Conversely, in the non-tabular setting, the development of hardness theory has been limited to specific cases under restrictive assumptions, making it difficult to relate these measures to environments that are of broader interest to the reinforcement learning community.

This paper presents the following contributions as the first step toward the ambitious objective of designing a principled benchmark for non-tabular reinforcement learning. In Section 2, we review the theory of hardness in both tabular and non-tabular settings, analyzing the limitations of tabular measures when applied to non-tabular environments and highlighting promising research directions. Section 3 identifies essential features for a principled non-tabular benchmarking library and introduces `pharos`, a new library designed to meet these needs. `Pharos` enables computation of tabular hardness measures, performs policy evaluation, and supports extensive environment customization crucial for evaluating agents in complex RL scenarios. Finally, Section 4 presents a case study using tabular Q-learning and DQN. This study, enabled by `pharos`, demonstrates the limitations of tabular hardness metrics in non-tabular settings (as highlighted by Conserva & Rauber (2022)) and underscores the need for novel metrics specifically tailored to the complexities of non-tabular representations, such as images.

## 2 Hardness in Reinforcement Learning

Section 2.1 formally introduces the reinforcement learning setting. Section 2.2 reviews the tabular hardness metrics that can be computed in practice. Section 2.3 highlights the key challenges in deriving a practically useful theory of hardness in the non-tabular setting. Section 2.4 formalizes the open problems and suggests future research directions.

### 2.1 Preliminaries

We formulate a reinforcement learning problem as a Markov decision process (MDP) $\mathcal{M} = (\mathcal{S}, \mathcal{A}, p, r, \gamma)$ composed of a set of states $\mathcal{S}$; a finite set of actions $\mathcal{A}$; a transition model $p : \mathcal{S} \times \mathcal{A} \times \mathcal{S} \to [0, 1]$ such that $\sum_{s'} p(s, a, s') = 1$ for every $(s, a) \in \mathcal{S} \times \mathcal{A}$; a reward function $r : \mathcal{S} \times \mathcal{A} \to \mathbb{R}$; and a discount factor $\gamma \in (0, 1)$. If for every $(s, a) \in \mathcal{S} \times \mathcal{A}$ there is a state $s' \in \mathcal{S}$ such that $p(s, a, s') = 1$, we say that the transition model $p$ is deterministic.

The interaction between the agent and the environment starts at time step $t = 0$ and ends at time $t = T$. For any $t \geq 0$, the agent selects an action $a_t$ from its policy $\pi : \mathcal{S} \to \mathcal{A}$ and the environment draws reward $r_{t+1}$ and next state $s_{t+1}$. For the episodic setting, we refer to the episode length as $H$.

The objective of a reinforcement learning agent is to find an optimal policy $\pi^*$ that maximizes the expected discounted return defined as $G_t^\pi = \mathbb{E}_\pi \left[ \sum_{k=0}^\infty \gamma^k r(s_t, \pi(s_t)) \right]$. The state-value function $V^\pi(s) = \mathbb{E}_\pi[G_t | S_t = s]$ quantifies the expected return when starting in state $s$ and following policy $\pi$. Similarly, the action-value function $Q^\pi(s, a) = \mathbb{E}_\pi[G_t | S_t = s, A_t = a]$ quantifies the expected return when starting in state $s$, taking action $a$, and then following policy $\pi$. The optimal state-value and action-value functions are denoted by $V^*$ and $Q^*$, respectively.

## 2.2 Tabular

In the tabular setting, the agent can maintain distinct value estimates for each state-action pair, facilitating systematic exploration. However, this also necessitates visiting a substantial portion of the state-action space, as the value of one state-action pair $(s, a)$ provides no direct information about the value of another $(s', a')$. For example, in a tree-structured episodic MDP, if $s$ and $s'$ are leaf nodes, their values are independent. While complete exploration is not strictly required due to bounded rewards and the possibility of pruning based on maximum return, the theoretical lower bound on regret (e.g., $\sqrt{HSAT}$ in the episodic case (Dann & Brunskill, 2015)) reflects the dependence on the number of states and actions.

**Hardness characterization.** To study the hardness of this setting, MDP complexity has been broken down into two categories: **visitation complexity** and **estimation complexity** (Conserva & Rauber, 2022). Visitation complexity refers to the difficulty an agent faces in visiting all states within the state space, which is crucial for learning an effective policy. Estimation complexity, on the other hand, deals with the challenges of estimating the optimal policy accurately based on the samples gathered, highlighting the discrepancy between the true optimal policy and the best policy derived from the given estimates of the transition and reward kernels. These two categories are complementary; visitation complexity focuses on the effort required to explore the state space thoroughly, while estimation complexity concerns the precision needed in policy estimation given the gathered data.

**Diameter.** The diameter of an MDP provides a measure of its visitation complexity, quantifying the difficulty of moving between states. Formally, the diameter D is defined as:

$$D := \sup_{s' \in \mathcal{S}} \inf_{\pi} T^{\pi}_{s \to s'}, \tag{1}$$

where represents the expected number of steps to reach state $s$ from $s'$ under policy $\pi$ (Jaksch et al., 2010). While the diameter effectively captures the challenge of gathering samples from the environment, it fails to account for the reward structure and the resulting value functions. Consequently, it doesn't address the estimation complexity of the MDP.

**Suboptimality gaps.** The sum of the reciprocals of the suboptimality gaps offers a measure of estimation complexity (Simchowitz & Jamieson, 2019). These gaps are defined as:

$$\Delta(s, a) := V^*(s) - Q^*(s, a) \tag{2}$$

Since $V^*(s) = \max_a Q^*(s, a)$ for all states $s$, the suboptimality gap $\Delta(s, a)$ quantifies the difference in expected return between taking the optimal action and taking action $a$ in state $s$. Intuitively, a larger gap $\Delta(s, a')$ for a suboptimal action $a'$ in state $s$ makes it easier to identify $a'$ as suboptimal. Unlike the diameter, this measure does not consider the difficulty of potentially hard-to-reach states, and, instead, it focuses on the complexity of accurately estimating the optimal policy.

## 2.3 Non-tabular

Contrary to the tabular setting, in non-tabular MDPs, the value function of state action pairs can be related even if state $s$ is not a predecessor of state $s'$ or vice versa. For example, in the Atari game Freeway (where the agent needs to cross a road), if the agent learns that, in order to maximize the total score, it needs to avoid being hit by a car, then it can transfer this knowledge to states that are not related from a tabular point of view, e.g. cars from different lanes. From a theoretical perspective, this relationship is formalized by assuming the functional form of the transition and reward functions.

**Feature dimension.** A well-studied non-tabular MDP model is the Low-Rank MDP (Zanette & Brunskill, 2019; Agarwal et al., 2020), which assumes a linear functional form in the reward and transition dynamics. Formally, there exists a known feature map $\phi : \mathcal{S} \times \mathcal{A} \to \mathbb{R}^d$ and parameters $\theta^r \in \mathbb{R}^d$, $\theta^P \in \mathbb{R}^d$ such that

$$r(s, a) = \phi(s, a)^T \theta^r \quad \text{and} \quad p(s, a, s') = \phi(s, a)^T \psi(s'). \tag{3}$$

We are also guaranteed that for every policy $\pi$ there exists a vector $\theta^\pi \in \mathbb{R}^d$ such that,

$$Q^\pi(s, a) = \phi(s, a)^T \theta^\pi. \tag{4}$$

While this setting is equivalent to the tabular one when $d = \|S\|\|A\|$ and the feature map is the indicator function for state-action pairs, for any $d < \|S\|\|A\|$, a certain degree of transfer between state-action pairs becomes possible, with lower dimensionality leading to higher levels of transfer. Although it is evident that the dimensionality of the feature map plays a crucial role in characterizing the hardness of the MDP, it becomes more challenging in this case to disentangle visitation complexity from estimation complexity, as can be done in the tabular setting. A reduction in the dimensionality of the feature map both decreases the number of state-action pairs the agent needs to visit—since there is greater transfer of information between pairs—and reduces the complexity of value function estimation, as fewer parameters are required to be learned. While the connection to linear regression theory offers theoretical advantages, this measure's practical application is severely limited in non-tabular settings as known feature maps that fit such strong assumptions are not available.

**Eluder dimension.** Alternative approaches to quantifying the complexity of this setting can leverage the framework based on the eluder dimension introduced by Osband & Van Roy (2014). The eluder dimension is generally defined as the longest possible sequence of tuples $(x_t, y_t)$ in a set such that it is not possible to estimate the function $x \mapsto y$ confidently. Intuitively, this measure quantifies the ability of a functional class to fit an arbitrary sequence, the longer the sequence the harder it is to estimate the functional form, and it can be thought of as the reinforcement learning counterpart of the Vapnik–Chervonenkis dimension. This measure has been further developed into the Bellman eluder dimension (Jin et al., 2021) and the generalized rank (Li et al., 2022). The eluder dimension could be particularly effective at capturing environment hardness because it both quantifies the visitation and estimation complexity. However, this measure can be exponentially large even for a shallow neural network with ReLU activation (Dong et al., 2021), meaning that the commonly used environments would all be assigned an infinite hardness value.

**Decision-estimation coefficient.** A more general measure of hardness that can be applied to any sequential decision-making setting (e.g., bandits) is the decision-estimation coefficient (Foster et al., 2021). Intuitively, the measure is the value of a game where the player must find a policy such that the regret is balanced with the estimation error for a worst-case problem instance. While the measure may be finite for environments of interest, it is not tractable to compute in practice.

## 2.4 Open problems

**Intractability of non-tabular measures.** The intractability of these hardness measures for relevant environments poses a key challenge for a practically useful theory of non-tabular hardness. There are two potential directions forward. The first is to develop measures that are less general and more instance-dependent. Specific tailoring of them to structure assumptions of the environment of interests in the reinforcement learning community may allow the development of tractable measures. The second one is to build practical approximations of the measures above. For example, while an exact feature map of a linear MDP is not available, it may be possible to build neural surrogates similar to those developed for climatology (Kochkov et al., 2024) and physics (Eghbalian et al., 2023).

**Representation learning.** While not specifically addressed in the theory, the hardness of representation learning in non-tabular reinforcement learning is well renowned. It is particularly clear to reinforcement learning researchers that an in-depth understanding of the representation complexity can lead to significantly better-performing agents. For example, Dabney et al. (2021); Lyle et al. (2022) shows that preventing feature collapse is very important for sparse-reward environments. Also, Nikishin et al. (2022); D'Oro et al. (2022); Schwarzer et al. (2023) have shown that one key factor to improve the sample efficiency of reinforcement learning agents is to avoid the representation collapse of the Deep Q-Network (DQN) (Mnih, 2013) by resetting the network with random weights. This is connected with feature collapse as explained by Ni et al. (2024). They show that several empirically validated techniques, e.g. stop-gradient and auxiliary loss

functions, are effectively avoiding feature collapse, and so allow the policy of the agents to collapse to a sub-optimal local minimum. A promising future direction can be the development of non-tabular measures of hardness specifically suited to capture the *representation complexity*, which could complement the visitation and estimation complexity quantified by tabular measures. However, it is important to note that different neural network architectures induce different inductive biases (Battaglia et al., 2018).

## 3    Benchmarking Reinforcement Learning

Section 3.1 proposes a set of desiderata that a principled benchmarking library for non-tabular reinforcement learning should possess. Section 3.2 reviews the currently available benchmarking libraries highlighting gaps and limitations. Finally, Section 3.3 introduces `pharos`, a novel benchmarking library designed to address these shortcomings.

### 3.1    Library desiderata

Creating a theoretically principled reinforcement learning benchmark is particularly challenging. Environments should contain a diverse set of challenges, be large-scale enough to generate enthusiasm within the research community and provide meaningful insights into the agents' strengths and weaknesses. We capture these requirements based on the five main features described below.

**Environment hardness.**    To create a theoretically principled benchmark, it's essential to have a methodology for quantifying the hardness of environments. At present, the only available measures are tabular, and while they may not fully reflect the complexity of non-tabular reinforcement learning, any new and potentially better measure will need to be evaluated against them. While computing them in small-scale environments is relatively simple, their complexity grows polynomially; for example, the diameter computational complexity, excluding logarithmic factors, is $\tilde{O}(|\mathcal{S}|^{3.5}|\mathcal{A}|)$, This introduces significant engineering challenges when scaling these measures to large-scale environments.

**Integration of agent training.**    Access to state-of-the-art reinforcement learning agents is of paramount importance when developing a novel algorithm, to compare against existing ones, or a novel hardness measure, to evaluate against agents' performances, which are the main use cases of a principled benchmarking library. To accommodate the needs of those different types of research, the library for the agent training should be inclusive for users with different levels of reinforcement learning agent training experience. In practice, critical components include the presence of user guides, tutorials, and a comprehensively documented codebase.

**Environment scale.**    While small-scale environments are useful for early prototyping, they are insufficient for fully showcasing an agent's capabilities. If a benchmarking library does not include sufficiently large environments, researchers are less likely to invest the time and effort required to set up their code and experiments. Benchmarking libraries often either consist of only tabular problems, only relatively simple non-tabular environments, or only complex visual non-tabular environments. This means that throughout algorithm development researchers either have to integrate various libraries into their code-base or are required to implement their own variants and wrappers. Ideally, the library should support researchers throughout their entire development process, from initial small-scale tests to large-scale evaluations.

**Environments instances.**    To expose the agent to a diverse set of challenges, it is desirable to be able to select different environment classes (e.g. both grid worlds and Atari games) but also different environment variations (e.g. variable sizes of a grid world). While most of the environment libraries are focused on providing many different environment classes, only a few allow the user to customize the environment instances (see Table 1). However, it can be vital to understand which component of an environment drives which aspect of hardness. For example, increasing the size of a grid world increases the visitation challenge, as it takes more time for the agent to reach farther states. However, it does not increase the estimation challenge, as the value function structure remains almost unvaried. While less crucial for agent benchmarking, this aspect is also important during the development of reinforcement agents since it enables researchers to gradually introduce more complexity to the environment, making it convenient to scale the applicability

Table 1: Comparison of hardness-focused reinforcement learning benchmarking libraries.

| Feature | bsuite | colosseum | bridge | pharos (Ours) |
|---|---|---|---|---|
| **Hardness measures** | Expert | Tabular | Tabular | Tabular |
| **Agent training** | dopamine | acme | SB3 | SB3 |
| **Environments scale** | Small | Small | Small/Large | Small/Large |
| **Policy evaluation** | Unavailable | Unavailable | Partial | Supported |
| **Custom instances** | Unavailable | Supported | Unavailable | Supported |
| **Custom observations** | Unavailable | Supported | Unavailable | Supported |

of algorithms from small- to large-scale settings. For instance, reducing the number of cars or lanes in the Atari game Freeway environment would reduce the challenge while maintaining the same core dynamics and hard-exploration challenge. This would enable researchers to gradually introduce more complexity to their algorithms as required by the increased complexity of the environment.

**Environments observations.** As mentioned in Section 2, representation learning is a key challenge in non-tabular reinforcement learning, with representation collapse being a potential obstacle to successfully learning the optimal policy. Currently, researchers in this area need to develop creative approaches to validate their results effectively. For example, Anand et al. (2019) proposed a novel representation learning technique, and, in order to validate it, they manually extracted the location of the agent and other features from the Atari RAM by investigating the C++ source code of Atari 2600 games (Whalen & Taylor, 2008; Engelhardt, 2019; Jentzsch & CPUWIZ, 2019). A library that natively supports the customization of the environment representations would significantly facilitate those lines of research. In addition to the validation, such flexibility would also allow for further study of the role of inductive biases induced by different neural network architectural choices in reinforcement learning (Hessel et al., 2019).

## 3.2 Related works

**Bsuite** (Osband et al., 2020) offers a benchmark suite based on expert intuitions about environment difficulty. While its simplicity and interpretability are valuable for early-stage experimentation, its limited scope and lack of complex, high-dimensional tasks hinder its ability to assess agent scalability to real-world problems. Furthermore, its reliance on expert judgment precludes environment creation or customization. Finally, its integration with the *dopamine* (Castro et al., 2018) framework, while powerful, caters more to experienced RL users than the broader community.

**Colosseum** (Conserva & Rauber, 2022) leverages tabular hardness metrics to provide a more theoretically grounded approach to hardness quantification. Although it supports non-tabular agents and representations, its primary focus remains on tabular settings. A key limitation is scalability: its environment suite is restricted to small-scale tasks, as computing hardness measures for larger environments becomes computationally prohibitive. While **Colosseum** integrates with the highly scalable **acme** (Hoffman et al., 2020) agent library, the discontinued development since October 2023 poses a potential concern for long-term support and compatibility with evolving RL research.

**Bridge** (Laidlaw et al., 2023) offers a suite of non-tabular environments with simplified, tabular representations, leveraging Julia's performance advantages for building the representations. These environments, including simplified Atari games using high frameskip, are designed to provide insights into environment hardness. While Python bindings facilitate agent interaction, the underlying Julia implementation can pose interoperability challenges for customization and extension. Furthermore, **Bridge** lacks scalable built-in analysis tools, requiring researchers to rely on external methods for thorough evaluation. **Bridge** integrates with Stable Baselines3 (Raffin et al., 2021), a popular and actively developed Python library containing state-of-the-art RL agents.

### 3.3 Pharos

`Pharos` is a novel benchmarking library that was specifically designed to satisfy the needs of researchers developing novel reinforcement learning agents and novel hardness measures.

**Capabilities.** Designed for principled benchmarking, `pharos` enables the computation of tabular hardness measures and the associated value functions. It integrates Stable Baselines3 for agent training and natively supports common stochastic environment types like sticky actions (Machado et al., 2018) and action randomization (Conserva & Rauber, 2022). Sticky actions introduce a non-zero probability of repeating the previous action, while action randomization resembles an environment-enforced epsilon-greedy exploration strategy. Computing value functions under sticky actions or action randomization are non-trivial, as the state value becomes dependent on the previous action. Further details on these challenges are provided in Appendix A.

**Scalability.** `Pharos` constructs environments by building scalable tabular representations, easily supporting millions of states. While conceptually straightforward, building such large state spaces presents significant scalability challenges. The underlying algorithm performs a depth-first search, expanding the state space using the transition function from an initial state. The primary challenge lies in efficiently tracking visited states to avoid loops, as the state space can rapidly exceed memory capacity. For instance, representing states as 30-integer tuples (approximately 300 bytes each in Python) requires 30GB of memory for a million states. Further details and the algorithm are provided in Appendix A and Algorithm 1.

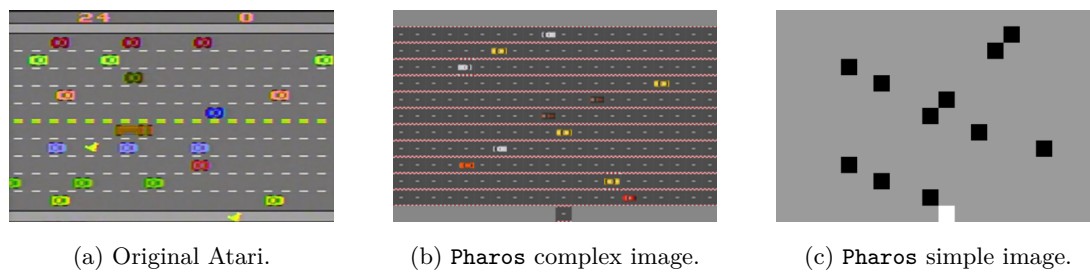

(a) Original Atari.      (b) `Pharos` complex image.      (c) `Pharos` simple image.

Figure 1: Atari like freeway available in `pharos` with different image representations.

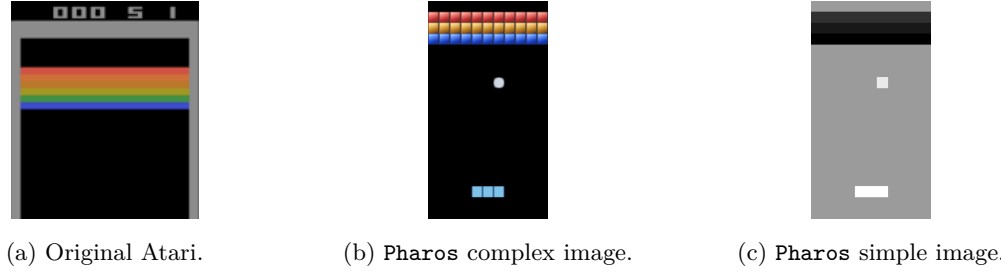

(a) Original Atari.      (b) `Pharos` complex image.      (c) `Pharos` simple image.

Figure 2: Atari like breakout available in `pharos` with different image representations.

**Available environments classes.** The `simple_grid` class of environments can be used to create grid world instances with different heights and widths. It is also possible to add reward and penalty locations. While the size of the class of environments grows unbounded with the height and width parameters, grid world environments are usually testbeds for small-scale testing of reinforcement learning agents. Two specific grid world instances of different sizes are provided as `frozen_lake` class of environments based on the widely used homonym environment. While the scale of these environments is usually small, they can provide a significant challenge due to the penalty locations. The `freeway` class of environments can be customized by modifying the speed at which the agent moves, the speed of the cars, the number of cars, and the length of lanes. It is also possible, differently from the standard Atari game, to enforce that the agent restarts from

the initial position at the bottom of the screen after hitting a car or to add a penalty for hitting a car. This significantly increases the exploration challenge. The `breakout` class of environments can be customized by changing the height, the number of columns, the number of rows with bricks, and the paddle size. For each environment, we provide image representations and a vector representation which fully describes the state of the environment (similar to the RAM in Atari). The image representation for freeway and breakout are respectively shown in Figure 1 and Figure 2, along with a comparison with the original Atari game representation.

## 4  Experiments

In Section 4.1 we examine the performance of the Q-learning algorithm in its tabular and non-tabular versions on small and large-scale environments to investigate how hard a simple and hard environment can be for a non-tabular and tabular agent, respectively. In Section 4.2, we analyse the results of training DQN agents with varying representations to provide an example of the inductive biases of neural network. In Section 4.3 we perform a correlation study with several linear models to investigate whether or not tabular hardness measures can capture non-tabular hardness.

**Experimental setup.**  A simple grid world and frozen lake are chosen as small-scale environment classes (less than 100 states), whereas freeway and breakout are chosen as large-scale environment classes (millions of states). Different environment instances are randomly chosen for each class, e.g., a different grid size for the grid worlds or a different number of lanes for `freeway`, to ensure a diverse set of challenges. For each environment instance, the DQN agent is trained with a simple image-based representation and a simple vector-based representation, and the results are averaged across five seeds. Running the same experiments with different image representations is left for future work. The hyperparameters of both the tabular and the non-tabular agents have been optimized for each environment class. The number of environment steps is $50k$ for the small-scale environment and $600k$ for the large-scale. For the vector representation, we have found that in practice the agent performs better when the values are re-scaled in the range $[0, 1]$ instead of raw values.

**Limitations.**  This study focused on a limited set of environments and a single agent to allow for an in-depth qualitative analysis of the results. Future work could expand the scope of this research by including a wider range of agents, environments, and representation types, which would further validate the findings and enhance the generalizability of the results. To partially address this limitation, we have run the same setup for a PPO agent and found that the performances of the two agents are high correlated (0.81) Tables describing the full results for the two agents are available in Appendix C.

### 4.1  Tabular vs non-tabular performances

**Motivation.**  As explained in Section 2, there are significant structural differences between the tabular and the non-tabular settings from a theoretical perspective. In this section, we show how these differences play out in practice by comparing the performance of Q-learning in small and large-scale environments in its tabular version (Watkins & Dayan, 1992) and in its most widespread q-learning inspired agent DQN (Mnih, 2013).

**Results.**  Figures 3 report the agent's performance in terms of cumulative regret, meaning that lower is better. It is evident that in the smaller environments, the tabular agent performs significantly better than its non-tabular counterpart, where the DQN agent is not able to find the optimal policy for the frozen lake when provided with an image-based representation. Instead, the non-tabular agent is better than the tabular counterpart for larger environments, with the tabular Q-learning not solving `breakout` in the given time limit.

**Discussion.**  While it is unsurprising that the tabular agent performs better in smaller environments than in larger ones and that its non-tabular counterpart struggles more with smaller environments, this distinction is crucial when discussing the applicability of tabular hardness measures in non-tabular settings. The clear

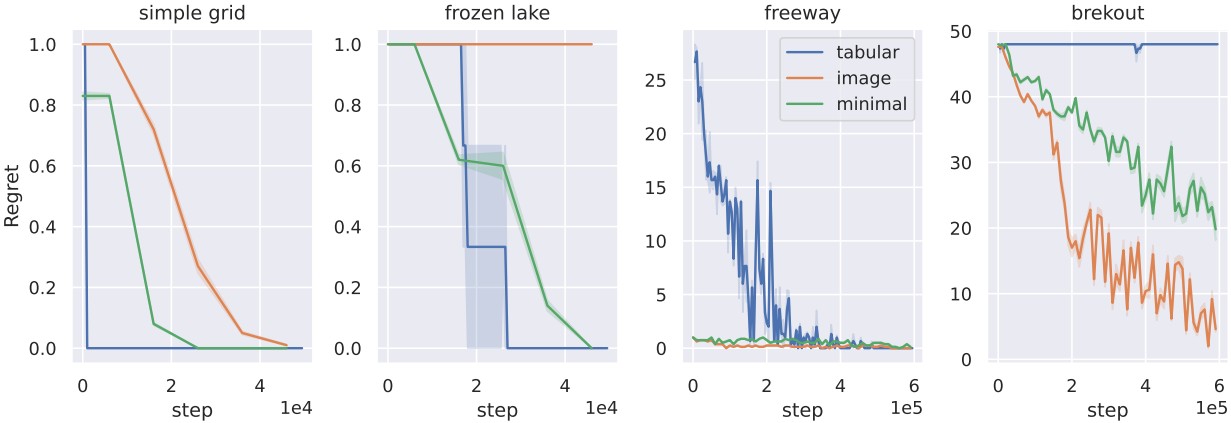

Figure 3: Regret plot for different environments available in pharos for the tabular case, for the case with a simple vector representation, and for the case of an image representation.

differences in what constitutes a difficulty for tabular versus non-tabular agents strongly suggest that these two settings are fundamentally distinct, making it unlikely that hardness measures from one can be reliably applied to the other.

## 4.2 Representation learning challenge

**Motivation** As mentioned in Section 2, different neural networks have different inductive biases. To support this claim in practice, we compare the regret incurred by the DQN agent in the same environment instances when an image observation and a vector observation were provided. The only difference in the agent is the feature extractor component, which is a convolutional neural network for the images and a fully connected neural network for vector observations.

**Results.** As shown in Figure 4, the type of representation significantly changes the incurred regret. However, we can notice that for the small-scale environments, `frozen_lake` and more evidently `simple_grid`, the regret when the agent is presented with an image is higher compared to a vector (points below the dotted line). The opposite instead holds for the large-scale environment `freeway` and `breakout` (points above the dotted line).

**Discussion.** While there are no currently available tools to effectively pinpoint a potential explanation of the presented results. This may be in line with the fact that neural networks, with their natural bias of producing smooth function, are not suited to be directly trained on tabular data (Beyazit et al., 2024) such as the one that is the vector encoding the state of the environment. While for `simple_grid` and `freeway`, the state representation is a vector containing the x y coordinates, the state representation of `freeway` and `breakout` needs to contain many different objects resulting in a longer and tabular like representation.

## 4.3 Correlation between hardness measures and non-tabular agents' performances

To better understand how well hardness measures capture the difficulty of environments, we fit linear models to predict the cumulative regret experienced by a DQN agent using these measures. We choose to limit ourselves to linear models since they are highly interpretable.

Although a single linear model applied across all environment instances (Section 4.3.1) does not yield an adequate fit, models tailored to specific representation types (Section 4.3.2) and environment classes (Section 4.3.3) show improved performance.

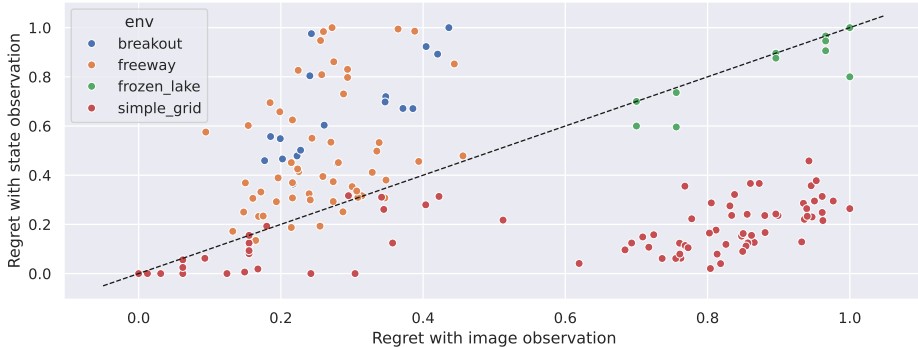

Figure 4: Normalized regret of DQN when provided with an image observation (x-axis) and with a state as observation (y-axis). DQN in breakout and freeway consistently incur more regret with a state observation compared with the image observation (points above the dotted line) whereas for a simple grid, the agent incurs more regret when provided with an image (points below the dotted line). For frozen lake, the regret is consistently high for both observation types.

### 4.3.1 Single model

The most general linear model is defined as,

$$
\begin{aligned}
\text{Regret.DQN} \sim{}& \alpha + \beta_1 \text{representation} + \beta_2 \text{breakout} + \beta_3 \text{freeway} + \beta_4 \text{frozen.lake} \\
& + \beta_5 \log \text{effective.horizon} + \beta_6 \log \text{sub.gaps} + \beta_7 \log \text{diameter},
\end{aligned}
\tag{5}
$$

where the predictions are representation, which is a dummy variable encoding whether the observation type (image or vector) provided to the agent, breakout, freeway, and frozen.lake, which are dummy variables encoding the environment class [1], effective.horizon, sub.gaps, and diameter, which are the tabular measure of hardness.

The $R^2$ score of this model is 0.09, and the fitted vs actual plot is shown in Figure 5. The poor fit of this model and the absence of any statistical significance in the model coefficient indicate that tabular hardness measures are not able to capture the hardness of the non-tabular task in a way that generalizes across environment classes and representation types.

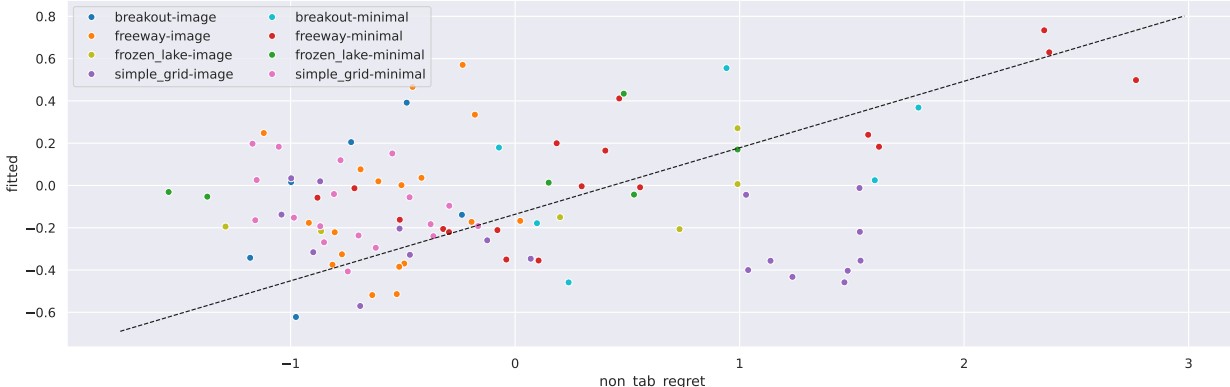

Figure 5: *Fitted vs actual plot* for the single model (Equation 5) where the non-tabular agent regret is inferred from tabular measure of hardness. The model is not able to fit well as shown by the fact that the points are not close to the identity diagonal line.

---

[1] We omit explicitly adding a dummy variable for the simple grid class as it is already implicitly captured in the constant $\alpha$.

### 4.3.2 Representation specific models

Given that the tabular measures can not account for the representation complexity of the observation, it may be the case that a better fit is obtained when fitting two separate models for each representation type. The linear model for this setting simplifies as,

$$
\begin{aligned}
\text{Regret.DQN}^{(\text{rep})} \sim {} & \alpha + \beta_2^{(\text{rep})}\text{breakout} + \beta_3^{(\text{rep})}\text{freeway} + \beta_4^{(\text{rep})}\text{frozen.lake} \\
& + \beta_5^{(\text{rep})}\log\text{effective.horizon} + \beta_6^{(\text{rep})}\log\text{sub.gaps} + \beta_7^{(\text{rep})}\log\text{diameter},
\end{aligned}
\tag{6}
$$

where we drop the dummy variable encoding the representation.

Differently from the previous model, where the effect of the representation type is only captured by the $\beta_1$ coefficient, in the model described by Equation 6 the two representation types have specific coefficients. In other words, while the number of parameters of the linear models decreases by one, the overall number of fitted parameters increases from 8 to 14.

The two settings are clearly different, as highlighted by the contrasting $R^2$ scores for the two models, with fitted vs. actual plots shown in Figure 6. In the case of **image-based representations**, the model achieves an $R^2$ score of 0.3, and none of the coefficients is statistically significant, indicating a weak relationship between the hardness measures and the agent's performance. In contrast, for **vector-based representations**, the model shows a higher $R^2$ score of 0.6, with all coefficients statistically significant except for the one corresponding to log effective.horizon.

While it is not always the case that an image observation makes the environment more complex than a vector representation, the difference in the fit of the linear models indicates the presence of a difference between the two representation types. The hardness induced by vector representations more closely aligns with tabular one whereas image observation induces a representation complexity not captured by the tabular measures.

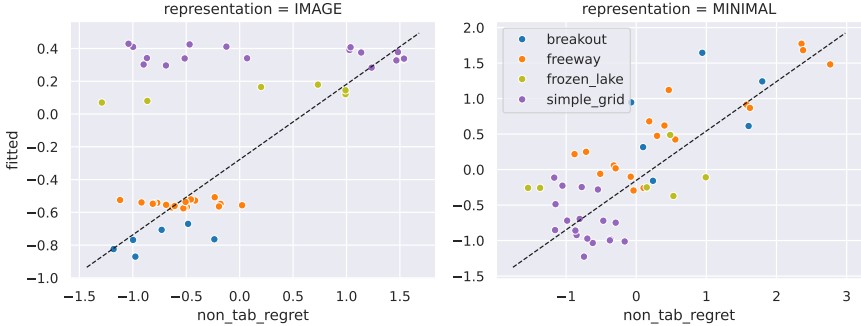

Figure 6: *Fitted vs actual plot* for models separately fitted on the different representation types (Equation 6). While the fit for the vector representation is decent, the model is clearly not able to capture the case of image representations.

### 4.3.3 Environment class specific models

Finally, we investigate fitting linear models for each environment class to understand whether the hardness measures are more robust to the representation type in individual classes. The models are in the form,

$$
\begin{aligned}
\text{Regret.DQN}^{(\text{env})} \sim {} & \alpha + \beta_1^{(\text{rep})}\text{representation} \\
& + \beta_2^{(\text{env})}\log\text{effective.horizon} + \beta_3^{(\text{env})}\log\text{sub.gaps} + \beta_4^{(\text{env})}\log\text{diameter},
\end{aligned}
\tag{7}
$$

with a total number of parameters equal to 20.

The $R^2$ score for `frozen_lake` is 0.96. The coefficients of the hardness measures are statistically significant while the dummy for the representation is not, meaning that, for this specific class of environments, the

hardness measures are robust to the two different representation types, and they are all important predictors of the agent's regrets.

The $R^2$ score is similarly high for `breakout` (0.92), but in this case, the representation and the log sub gaps are the ones with statistically significant coefficients. This means that for the breakout class, the types of representation significantly influence the regret of the agent, and, among the hardness measures, the sub gap is the one that is better suited to capture the hardness.

The $R^2$ score for `simple_grid` is 0.42 with representation dummy and log diameter being statistically significant. While this indicates that the representation and the diameter play an important role in identifying the hardness of this class of environment, the model overestimates the hardness of a simple grid environment without penalty locations (shown as a cluster in the upper left corner of Fig 7). This is in line with the fact that the presence/absence of penalty position is not captured by the diameter since, by definition, the diameter doesn't include any term from the reward function. While the sub-gap could potentially capture this, it does not do so sufficiently, and it ends up not as it is not statistically significant.

The $R^2$ score for `freeway` is 0.67, with only the representation having a statistically significant coefficient. This means that for this class of environments, the hardness measures are not able to capture the hardness. The representation plays an important role in this class. This is in line with the fact that the freeway is a hard exploration environment, and image representation is more prone to induce representation collapse, which is not captured by the tabular measures of hardness.

The overall $R^2$ scores are 0.08 for the single model (Equation 5), 0.48 when combining the separate models for representation types (Equation 6), and 0.65 for the separate models for environments classes (Equation 7).

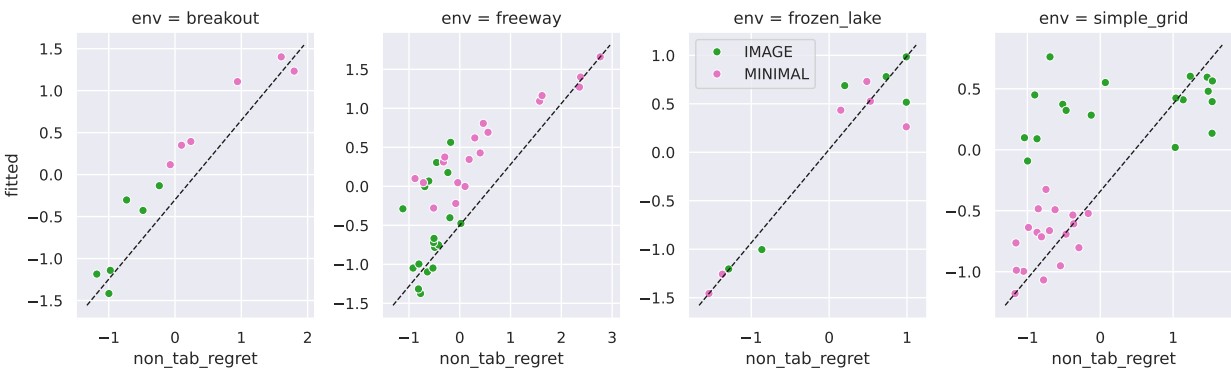

Figure 7: *Fitted vs actual plot* for models separately fitted on the different environment classes. There is a relatively good fit between the classes with the exception of simple.grid.

## 5 Conclusion

This work contributes a significant step towards principled benchmarking of non-tabular reinforcement learning agents. Our thorough literature review reveals a critical gap between the theory and practice of hardness measures in this setting. Readily computable measures exist for tabular environments, but practical counterparts are lacking in the non-tabular domain. Crucially, no available measures account for the impact of different state representations. Consequently, identical environments presented with varying representations, such as images or vectors, are assigned the same hardness value, despite posing different learning challenges.

Our case study, uniquely enabled by `pharos`, employed tabular q-learning and DQN to expose the limitations of applying tabular hardness measures to non-tabular settings. A key limitation originates from the significant difference in experienced hardness when the same environment is encoded as tabular versus non-tabular. We demonstrated this in practice by showing that tabular Q-learning outperforms DQN in small-scale environments, while the opposite holds for larger ones. Further, tabular measures do not ac-

count for different representations, and we showed that varying the representation (vector versus image) significantly influenced DQN's performance. Finally, we showed that, for some environment classes, certain tabular measures correlate with DQN performance, suggesting their potential utility as building blocks for future, more sophisticated, representation-aware metrics. While this case study provides a valuable initial investigation, future work should expand this analysis to encompass a wider range of agents, environments, and potentially more complex, non-linear models to analyze the relationship between agent performance and hardness measures.

In summary, this work illuminates critical shortcomings in existing non-tabular benchmarking practices, identifies key areas for future research, and provides `pharos` as a powerful tool to facilitate progress in this important domain.

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

## A    Scalability

**State space builder.**    The state space builder requires a transition function that computes the next state for any state and given action, the termination function that determines whether a state is terminal, and the starting state. The `STATE` is a tuple of integers that fully describes the state of the environment. For example, the position of the agent, the ball, and the brick in breakout. The `ACTION` is an integer. The algorithm starts by populating a queue of states to query with the starting state. At every iteration, the first element is popped from the queue, and next states are computed for each action. Any next state that has not been previously visited is then (i) added to the queue, (ii) its value is stored in a matrix on disk where the row corresponds to the state index and the columns to the state representation dimensionality, (iii) the index of the next state is stored in another on-disk matrix where the rows correspond to the previous state index and the columns to the action, and (iv) its value is stored in a hash map that keeps track of all the previously visited states.

**Scalability challenges.**    The two key challenges in terms of memory are the queue, which can grow to a large extent when there are long sequences of novel states, and the hash map, which grows linearly with the number of states in the environment. In practice, we have found the maximum size of the queue does not go over thousands of states, meaning that this occupies a relatively low proportion of the memory. On the contrary, the increase of the size of the hash map is not avoidable, and, for large state space, it cannot be stored in memory. To provide a reference for the potential size of this set we can assume that the state is represented by a tuple of 30 integers, which occupies 300 bytes of memory in Python. For an environment with one hundred million states, this corresponds to more than 30 GB of memory. In order to scale to larger-than-memory state spaces, we therefore need an in-disk hash map that has high key retrieval speed and and high compression, with possibly slower IO speed. While slower IO speed can be compensated with in-memory buffers, the retrieval speed is essential because the algorithm is constantly querying whether a next state has been seen before, and the compression rate is also necessary to avoid that the environment occupy too much disk space. We have explored several options. `sqlitedict` was immediately excluded due to slow retrieval speed. `DiskDict` has a high key retrieval speed but does not natively support compression. `ZODB` is reasonably fast in key retrieval and supports compression but `RocksDB` is better. This library builds the hash map by storing sstables on disk.

**Reward function.**    After the state space has been built, the library goes on with the computation of the rewards. While the rewards could be computed at the same time as building the state space, in practice, we have found it better to do it after because rewards can be computed in parallel based on the stored transitions, and we want to avoid further straining the intensive IO happening while building the state space.

**Representation function.**    Similar to the rewards, the representation function of the environment can be computed after the state space has been fully created. Given that the representation are only required during the interaction with the agent (no metric is computed based on them) we have found more beneficial to compute them on the fly to reduce the on-disk space requirements. For an 64 by 64 image representation, the uncompressed space requirements for one hundred million states would be larger than 3 TB.

**Stochastic environments.**    Although incorporating support for state transition probability functions could be a future enhancement for the library, we believe that the substantial computational cost associated with this approach is not justified at present. Our assessment is based on the observation that most of the reinforcement learning community's work has focused on stochasticity types that can be derived from a deterministic transition function. While the computation of environment properties under action randomization is largely the same as in non-randomized settings, sticky action introduces a notable change to the environment's structure. Specifically, in a sticky action environment, the next state depends not only on the previous state and the current action but also on the action taken in the previous step. This added dependency increases the complexity of computing environment properties, such as the optimal value function, which now requires a matrix with dimensions corresponding to the number of states by the number of actions, rather than a single vector of state lengths. Given the previously described design, an important

---

**Algorithm 1** State Space Builder

---

1: **Input:** $starting\_state \leftarrow$ STATE             ▷ Initial state of the environment
2: **Input:** $transition\_function(s, action)$          ▷ Function to compute the next state
3: **Input:** $is\_terminal(s)$           ▷ Function to check if a state is terminal
4: $queue \leftarrow [starting\_state]$          ▷ Initialize queue with the starting state
5: $visited\_states \leftarrow \{\}$          ▷ Hash map to track visited states
6: $state\_index \leftarrow 0$          ▷ Index for the current state
7: **while** $queue$ is not empty **do**
8:     $current\_state \leftarrow$ pop($queue$)          ▷ Pop first state from the queue
9:     **if** $is\_terminal(current\_state)$ **then**
10:        **continue**          ▷ Skip if state is terminal
11:     **end if**
12:     **for each** $action$ **in** $ACTIONS$ **do**
13:        $next\_state \leftarrow transition\_function(current\_state, action)$      ▷ Compute next state
14:        **if** $next\_state$ **not in** $visited\_states$ **then**
15:           append($queue$, $next\_state$)          ▷ Add new state to queue
16:           store_state(state_index, $next\_state$)          ▷ Store state in disk matrix
17:           store_transition(state_index, action, $next\_state$)     ▷ Store transition in another matrix
18:           $visited\_states[next\_state] \leftarrow state\_index$       ▷ Mark state as visited with its index
19:           $state\_index \leftarrow state\_index + 1$       ▷ Update state index for the next state
20:        **end if**
21:     **end for**
22: **end while**

---

characteristic of the library is that the user can change the rewards and the representation without the necessity of building the full state space. This is particularly important for researchers that are studying the representation learning challenges of deep reinforcement learning.

## B  Environment representations

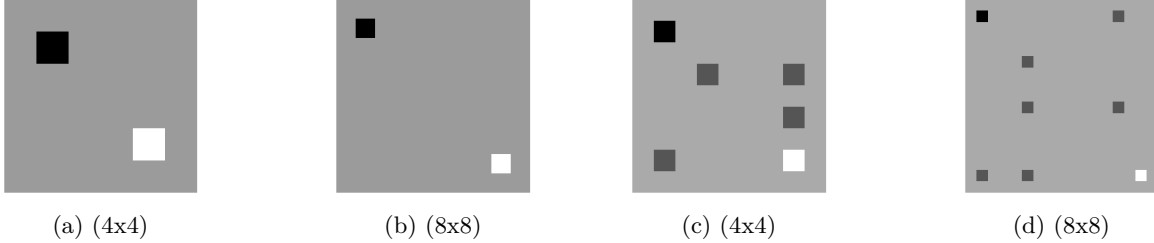

(a) (4x4)  (b) (8x8)  (c) (4x4)  (d) (8x8)

Figure 8: Two different instances of SimpleGrid(height, width) and FrozenLake(height, width). The agent (black) has the simple task of reaching the goal (white). Frozen lake include holes (gray) that the agent needs to avoid.

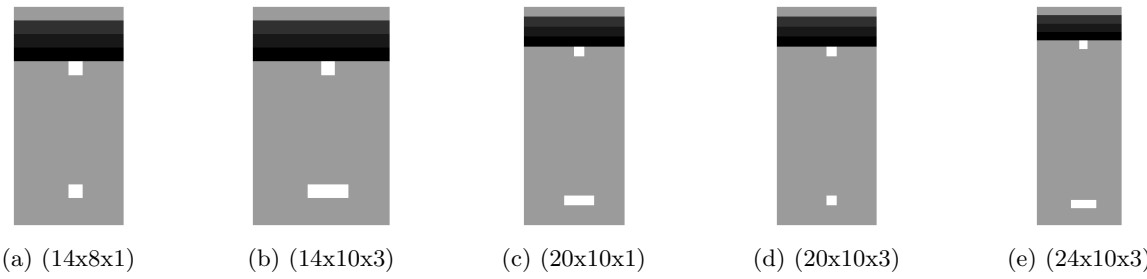

(a) (14x8x1)  (b) (14x10x3)  (c) (20x10x1)  (d) (20x10x3)  (e) (24x10x3)

Figure 9: Five different instances of Breakout(height, width, paddle_width). The black dots at the top of the figure represent the bricks that the agent needs to destroy by hitting the ball (white top dot) using the paddle (bottom dots).

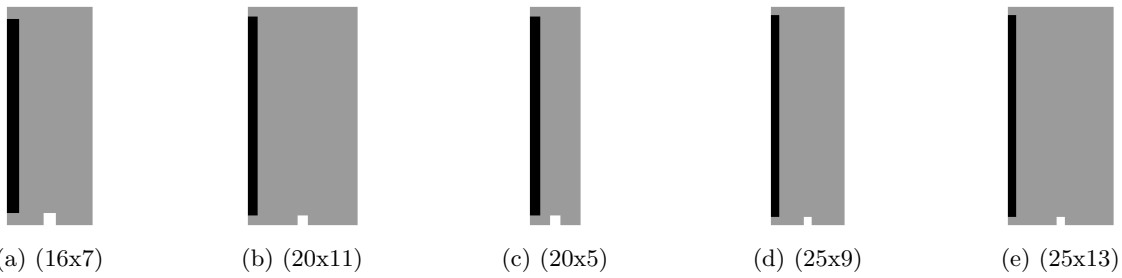

(a) (16x7)  (b) (20x11)  (c) (20x5)  (d) (25x9)  (e) (25x13)

Figure 10: Five different instances of Freeway(num_cars, lane_length). The black dots on the left side of the figure represent cars whereas the white dot on the bottom is the agent. The goal of the agent is to reach the top by moving vertically without being hit by the cars moving horizontally. Users can modify the number of cars and the length of the lanes. Increasing the number of cars or reducing the lane length will make the environment more difficult, as the agent must navigate a higher number of moving vehicles within a more restricted space.

## C  Agent's performances

Table 2: Cumulative return of DQN and PPO agents on `simple grid` environments.

| environment parameter | randomization | PPO IMAGE | PPO MINIMAL | DQN IMAGE | DQN MINIMAL |
|---|---|---|---|---|---|
| height=4, width=4 | None | $0.03 \pm 0.04$ | $0.78 \pm 0.20$ | $0.10 \pm 0.00$ | $1.00 \pm 0.00$ |
| | random | $0.24 \pm 0.17$ | $0.89 \pm 0.00$ | $0.14 \pm 0.00$ | $0.92 \pm 0.00$ |
| | stick | $0.00 \pm 0.01$ | $0.45 \pm 0.28$ | $0.22 \pm 0.00$ | $0.99 \pm 0.00$ |
| height=4,width=4 | None | $0.42 \pm 0.41$ | $0.86 \pm 0.19$ | $1.00 \pm 0.00$ | $1.00 \pm 0.00$ |
| | random | $0.94 \pm 0.09$ | $1.00 \pm 0.00$ | $1.00 \pm 0.00$ | $1.00 \pm 0.00$ |
| | stick | $0.10 \pm 0.07$ | $0.98 \pm 0.02$ | $1.00 \pm 0.00$ | $1.00 \pm 0.00$ |
| height=6,width=6 | None | $0.00 \pm 0.00$ | $0.57 \pm 0.41$ | $0.10 \pm 0.00$ | $0.99 \pm 0.00$ |
| | random | $0.86 \pm 0.00$ | $0.91 \pm 0.03$ | $0.11 \pm 0.00$ | $0.98 \pm 0.00$ |
| | stick | $0.03 \pm 0.03$ | $0.07 \pm 0.06$ | $0.17 \pm 0.00$ | $0.99 \pm 0.00$ |
| height=6,width=6 | None | $0.20 \pm 0.17$ | $0.97 \pm 0.01$ | $0.99 \pm 0.00$ | $1.00 \pm 0.00$ |
| | random | $0.97 \pm 0.02$ | $0.99 \pm 0.01$ | $1.00 \pm 0.00$ | $1.00 \pm 0.00$ |
| | stick | $0.15 \pm 0.14$ | $0.94 \pm 0.05$ | $0.99 \pm 0.00$ | $1.00 \pm 0.00$ |
| height=8,width=8 | None | $0.09 \pm 0.12$ | $0.44 \pm 0.36$ | $0.10 \pm 0.00$ | $0.99 \pm 0.00$ |
| | random | $0.83 \pm 0.01$ | $0.89 \pm 0.03$ | $0.10 \pm 0.00$ | $0.98 \pm 0.01$ |
| | stick | $0.00 \pm 0.00$ | $0.08 \pm 0.07$ | $0.12 \pm 0.00$ | $0.98 \pm 0.00$ |
| height=8,width=8 | None | $0.04 \pm 0.03$ | $0.66 \pm 0.38$ | $0.98 \pm 0.00$ | $0.99 \pm 0.00$ |
| | random | $0.85 \pm 0.01$ | $1.00 \pm 0.00$ | $0.99 \pm 0.00$ | $1.00 \pm 0.00$ |
| | stick | $0.00 \pm 0.00$ | $0.35 \pm 0.05$ | $0.99 \pm 0.00$ | $0.99 \pm 0.00$ |

Table 3: Cumulative return of DQN and PPO agents on `frozen lake` environments.

| environment parameter | randomization | PPO IMAGE | PPO MINIMAL | DQN IMAGE | DQN MINIMAL |
|---|---|---|---|---|---|
| height=4,width=4 | None | $0.00 \pm 0.00$ | $0.11 \pm 0.16$ | $0.00 \pm 0.00$ | $0.98 \pm 0.01$ |
| | random | $0.00 \pm 0.00$ | $0.64 \pm 0.03$ | $0.01 \pm 0.00$ | $0.71 \pm 0.01$ |
| | stick | $0.00 \pm 0.00$ | $0.59 \pm 0.04$ | $0.00 \pm 0.00$ | $0.98 \pm 0.00$ |
| height=8, width=8 | None | $0.00 \pm 0.00$ | $0.06 \pm 0.06$ | $0.00 \pm 0.00$ | $0.96 \pm 0.00$ |
| | random | $0.00 \pm 0.00$ | $0.72 \pm 0.09$ | $0.01 \pm 0.00$ | $0.89 \pm 0.01$ |
| | stick | $0.00 \pm 0.00$ | $0.30 \pm 0.29$ | $0.00 \pm 0.00$ | $0.94 \pm 0.01$ |

Table 4: Cumulative return of DQN and PPO agents on `freeway` environments.

| environment parameter | randomization | PPO IMAGE | PPO MINIMAL | DQN IMAGE | DQN MINIMAL |
|---|---|---|---|---|---|
| lane length=11 n cars=20 player speed=1 | None | $-2.77 \pm 2.16$ | $0.01 \pm 0.78$ | $0.12 \pm 0.10$ | $-0.60 \pm 0.60$ |
| | random | $-22.26 \pm 12.48$ | $-9.80 \pm 11.04$ | $-2.00 \pm 0.36$ | $-7.60 \pm 5.08$ |
| | stick | $-0.06 \pm 0.05$ | $-0.08 \pm 0.17$ | $-0.11 \pm 0.05$ | $-0.44 \pm 0.35$ |
| lane length=11 n cars=20 player speed=2 | None | $0.00 \pm 0.00$ | $0.65 \pm 0.08$ | $-0.65 \pm 0.48$ | $-1.32 \pm 0.11$ |
| | random | $-45.47 \pm 10.50$ | $-13.93 \pm 14.85$ | $-7.78 \pm 0.86$ | $-25.02 \pm 1.28$ |
| | stick | $-3.01 \pm 4.25$ | $-1.89 \pm 1.67$ | $-2.97 \pm 0.57$ | $-11.38 \pm 0.36$ |
| lane length=13 n cars=25 player speed=1 | None | $0.20 \pm 0.59$ | $-0.24 \pm 0.91$ | $-0.11 \pm 0.35$ | $-0.62 \pm 0.42$ |
| | random | $-12.30 \pm 11.27$ | $-4.68 \pm 4.69$ | $-1.60 \pm 0.54$ | $-2.63 \pm 0.74$ |
| | stick | $-0.20 \pm 0.25$ | $0.24 \pm 0.37$ | $-1.14 \pm 0.18$ | $-0.37 \pm 0.30$ |
| lane length=13 n cars=25 player speed=2 | None | $-5.99 \pm 4.94$ | $-0.48 \pm 1.04$ | $-0.29 \pm 0.27$ | $-1.84 \pm 0.27$ |
| | random | $-38.84 \pm 14.27$ | $-4.92 \pm 2.16$ | $-8.74 \pm 1.78$ | $-22.32 \pm 0.29$ |
| | stick | $-5.34 \pm 5.15$ | $-0.22 \pm 0.31$ | $-2.89 \pm 0.65$ | $-10.69 \pm 3.65$ |
| lane length=5 n cars=20 player speed=1 | None | $-0.80 \pm 1.13$ | $-0.01 \pm 0.01$ | $-0.75 \pm 0.42$ | $-4.82 \pm 0.56$ |
| | random | $-55.62 \pm 10.27$ | $-65.34 \pm 5.13$ | $-8.46 \pm 1.38$ | $-41.34 \pm 1.98$ |
| | stick | $0.00 \pm 0.00$ | $-1.00 \pm 0.83$ | $-6.13 \pm 1.05$ | $-32.46 \pm 0.85$ |
| lane length=5 n cars=20 player speed=2 | None | $-7.27 \pm 5.17$ | $-0.01 \pm 0.02$ | $-2.42 \pm 0.65$ | $-7.76 \pm 0.22$ |
| | random | $-122.50 \pm 29.65$ | $-106.40 \pm 4.32$ | $-56.41 \pm 1.56$ | $-83.72 \pm 2.16$ |
| | stick | $-16.59 \pm 22.98$ | $-0.00 \pm 0.00$ | $-18.41 \pm 5.79$ | $-44.41 \pm 2.51$ |
| lane length=7 n cars=16 player speed=1 | None | $-5.80 \pm 4.81$ | $0.24 \pm 0.55$ | $0.33 \pm 0.14$ | $-2.27 \pm 1.36$ |
| | random | $-2.15 \pm 1.58$ | $-25.99 \pm 17.00$ | $-3.09 \pm 0.89$ | $-1.40 \pm 0.27$ |
| | stick | $-4.92 \pm 6.95$ | $0.18 \pm 0.25$ | $-0.14 \pm 0.27$ | $-0.06 \pm 0.08$ |
| lane length=7 n cars=16 player speed=2 | None | $-4.01 \pm 3.88$ | $0.41 \pm 0.30$ | $-0.18 \pm 0.28$ | $-4.23 \pm 0.40$ |
| | random | $-37.76 \pm 7.44$ | $-11.99 \pm 12.33$ | $-8.72 \pm 1.40$ | $-6.45 \pm 1.20$ |
| | stick | $0.00 \pm 0.00$ | $-0.51 \pm 0.73$ | $-3.41 \pm 1.43$ | $-10.96 \pm 5.76$ |
| lane length=9 n cars=25 player speed=1 | None | $-4.12 \pm 2.95$ | $-1.48 \pm 1.25$ | $-0.51 \pm 0.28$ | $-1.94 \pm 0.19$ |
| | random | $-64.03 \pm 0.12$ | $-38.86 \pm 25.74$ | $-8.30 \pm 1.29$ | $-27.00 \pm 0.78$ |
| | stick | $0.00 \pm 0.00$ | $-1.19 \pm 1.69$ | $-2.50 \pm 1.04$ | $-6.22 \pm 1.20$ |
| lane length=9 n cars=25 player speed=2 | None | $-2.46 \pm 1.75$ | $-0.65 \pm 0.54$ | $-1.14 \pm 0.24$ | $-2.04 \pm 0.55$ |
| | random | $-76.75 \pm 20.22$ | $-38.05 \pm 37.94$ | $-20.54 \pm 5.08$ | $-34.51 \pm 4.82$ |
| | stick | $-26.97 \pm 2.22$ | $-11.89 \pm 12.85$ | $-7.37 \pm 2.00$ | $-25.16 \pm 1.47$ |

Table 5: Cumulative return of DQN and PPO agents on `breakout` environments.

| Environment parameter | Random | PPO IMAGE | PPO MINIMAL | DQN IMAGE | DQN MINIMAL |
|---|---|---|---|---|---|
| extra paddle width=0 height=12,columns=10 | None | $11.89 \pm 8.61$ | $3.50 \pm 1.64$ | $49.84 \pm 0.56$ | $34.43 \pm 2.91$ |
| | random | $5.27 \pm 0.21$ | $4.12 \pm 0.14$ | $11.17 \pm 0.43$ | $9.49 \pm 0.43$ |
| | stick | $2.92 \pm 2.75$ | $6.82 \pm 0.27$ | $15.71 \pm 0.93$ | $12.72 \pm 0.41$ |
| extra paddle width=0 height=12,columns=8 | None | $4.35 \pm 0.50$ | $9.04 \pm 4.02$ | $42.67 \pm 0.27$ | $35.09 \pm 1.17$ |
| | random | $7.74 \pm 0.92$ | $2.17 \pm 0.03$ | $15.52 \pm 0.84$ | $13.73 \pm 0.53$ |
| | stick | $5.67 \pm 2.37$ | $6.26 \pm 3.20$ | $21.67 \pm 1.42$ | $14.81 \pm 0.76$ |
| extra paddle width=0 height=16,columns=8 | None | $2.02 \pm 0.02$ | $2.00 \pm 0.00$ | $36.90 \pm 0.38$ | $25.80 \pm 0.64$ |
| | random | $5.73 \pm 0.26$ | $1.98 \pm 0.00$ | $11.80 \pm 0.10$ | $10.51 \pm 0.19$ |
| | stick | $2.05 \pm 0.08$ | $2.00 \pm 0.00$ | $15.65 \pm 0.76$ | $13.43 \pm 0.81$ |
| extra paddle width=0 height=20,columns=8 | None | $4.56 \pm 3.23$ | $7.99 \pm 6.70$ | $38.85 \pm 2.50$ | $19.15 \pm 2.33$ |
| | random | $3.03 \pm 1.23$ | $2.68 \pm 1.73$ | $11.83 \pm 0.43$ | $8.68 \pm 0.19$ |
| | stick | $1.47 \pm 2.08$ | $3.09 \pm 1.94$ | $17.15 \pm 0.36$ | $10.09 \pm 0.58$ |
| extra paddle width=0 height=24,columns=8 | None | $2.67 \pm 1.72$ | $2.03 \pm 2.61$ | $35.00 \pm 0.69$ | $20.67 \pm 4.46$ |
| | random | $3.78 \pm 0.43$ | $1.38 \pm 1.35$ | $10.89 \pm 0.72$ | $8.04 \pm 0.10$ |
| | stick | $2.62 \pm 1.86$ | $2.66 \pm 0.57$ | $14.89 \pm 0.37$ | $8.92 \pm 0.40$ |
| extra paddle width=1 height=12,columns=10 | None | $12.81 \pm 7.14$ | $14.46 \pm 8.27$ | $52.66 \pm 0.65$ | $38.64 \pm 1.93$ |
| | random | $11.92 \pm 0.29$ | $7.24 \pm 4.06$ | $25.78 \pm 0.32$ | $26.41 \pm 0.36$ |
| | stick | $12.94 \pm 1.91$ | $6.73 \pm 3.86$ | $29.25 \pm 1.71$ | $31.43 \pm 0.89$ |
| extra paddle width=1 height=12,columns=8 | None | $24.36 \pm 3.72$ | $13.51 \pm 4.34$ | $43.85 \pm 1.14$ | $39.84 \pm 0.66$ |
| | random | $16.13 \pm 0.27$ | $17.41 \pm 2.32$ | $28.83 \pm 0.86$ | $30.25 \pm 0.49$ |
| | stick | $19.74 \pm 1.24$ | $16.48 \pm 0.97$ | $33.99 \pm 0.77$ | $34.42 \pm 0.92$ |
| extra paddle width=1 height=16,columns=8 | None | $2.00 \pm 0.00$ | $2.00 \pm 0.00$ | $40.90 \pm 1.14$ | $33.70 \pm 2.46$ |
| | random | $7.41 \pm 1.47$ | $5.06 \pm 4.36$ | $23.48 \pm 0.82$ | $21.59 \pm 0.82$ |
| | stick | $3.57 \pm 2.23$ | $2.00 \pm 0.00$ | $30.50 \pm 0.57$ | $27.12 \pm 1.27$ |
| extra paddle width=1 height=20,columns=8 | None | $8.34 \pm 4.36$ | $4.00 \pm 0.00$ | $42.31 \pm 0.34$ | $32.36 \pm 1.46$ |
| | random | $8.98 \pm 0.58$ | $5.77 \pm 1.64$ | $24.54 \pm 0.76$ | $22.18 \pm 0.35$ |
| | stick | $10.38 \pm 0.71$ | $4.76 \pm 1.07$ | $30.06 \pm 1.32$ | $28.05 \pm 1.52$ |
| extra paddle width=1 height=24,columns=8 | None | $5.01 \pm 2.63$ | $14.54 \pm 11.14$ | $40.75 \pm 0.55$ | $31.28 \pm 1.77$ |
| | random | $7.88 \pm 0.67$ | $7.03 \pm 3.19$ | $24.50 \pm 0.86$ | $23.44 \pm 0.78$ |
| | stick | $2.27 \pm 0.39$ | $10.70 \pm 6.94$ | $28.81 \pm 0.72$ | $29.52 \pm 0.59$ |

