# OpenReview forum: "Towards Principled Benchmarking of Non-tabular Reinforcement learning"
_TMLR — Rejected by TMLR_

### Review · Reviewer_5nmb · 2024-11-26

**Summary Of Contributions:**

This paper summarized the existing results no evaluating the hardness of environments in both tabular and non-tabular cases. To facilitate research in RL, the authors proposed a new library Pharos that contains a range of environments that allow for customization, including simplified Atari games. The authors showed that using the library, some theoretically sound metrics in the tabular setting do not translate to the non-tabular case, and the regret incurred by DQN using different representations could differ drastically.

**Audience:**

Yes

**Broader Impact Concerns:**

Not applicable.

**Claims And Evidence:**

No

**Requested Changes:**

Please refer to the above for requested changes.

**Strengths And Weaknesses:**

**Strengths**:\
This paper studies an important topic. Hardness of environments if known, can guide the researchers to choose suitable environments that reflect the capability of the their agents. The authors have provided a helpful survey of related work and technical topics. The new customizable library is appreciated since it is often desired to evaluate different levels of difficulty under the same environment. The authors' empirical results that the tabular metrics do not work well in the non-tabular case are, though expected, helpful in calling for investigation into development of the more general case.

**Weakness**:\
My concern is that the paper is somewhat loosely connected and does not very well support "principled benchmarking" in the title. The authors used a linear fit to the DQN regret using a mixture of environments, tabular metrics and representations but did not explain why such mixture was reasonable. While using a linear model is most interpretable, without more complex models it seems unfair to state the following in Section 4.3.1
> The poor fit of this model and the absence of any statistical significance in the model coefficient indicate that tabular hardness measures are not able to capture the hardness of the non-tabular task in a way that generalizes across environment classes and representation types.

In Section 4.3.3 the authors used a refined linear model to conduct the analysis and concluded that some variables were significant in for a class of environments, for example Breakout.  This looks confusing to me as the authors did not define what is really a class of environments. How was the "breakout class" defined? Can the authors give more examples? I am expecting to see not just trivial definition of gridworlds with larger sizes, but actually a connection between different environments. If we had a reasonable definition for a class of Atari games, we could potentially use it to characterize the hardness even just empirically.

My another concern is that the paper revealed many issues of the existing protocols but did not propose a viable solution. As the authors put in the conclusion:
> we showed that, for some environment classes, certain tabular measures correlate with DQN performance, suggesting their potential utility as building blocks for future, more sophisticated, representation-aware metrics.

This makes the paper seems rather incomplete to me: the authors showed correlation but did not suggest an usable metric. IMO, the paper could benefit from comprehensive evaluation on not just linear but also higher order models on a selected subset of variables to help conclude what might be an interesting combination as metric.

Minor questions:
 * the simplified Atari games in Pharos look similar to the MinAtar library, what is their difference?\
MinAtar: An Atari-Inspired Testbed for Thorough and Reproducible Reinforcement Learning Experiments, Kenny Young and Tian Tian
 * the paper has several misspellings and undefined notations:
    - in eq. 3 the $\psi$ is undefined, maybe the authors meant $\theta^P$.
    - in the Eluder paragraph:  "the harder it is estimate";  "and it can be though"

---

> ### Author Response · Authors · 2024-12-15
> **Response (1/2)**
>
> Thanks for reading the paper, and for the insightful comments.
>
> ```
> My concern is that the paper is somewhat loosely connected and does not very well support "principled benchmarking" in the title. The authors used a linear fit to the DQN regret using a mixture of environments, tabular metrics and representations but did not explain why such mixture was reasonable. While using a linear model is most interpretable, without more complex models it seems unfair to state the following in Section 4.3.1
> ```
> As mentioned in the title, our work represent a step *towards* principled benchmarking, which, in the non-tabular setting, is particularly complex due to the absence of computationally feasible hardness measures and the necessity of deep neural networks.
>
> Thanks for rasising this point, we will add the following explanation to the revision:
>
> The reason why we chose a simple grid world and frozen lake is because grid worlds are the most widely used classes of small scale environmnets in the literature, e.g. the Wumpus world from [1].
> For large scale environment, breakout and freeway instead represent two emblematic environments from the Atari suite, with breakout being part of the original five games present in [2] and freeway representing a prototypical environment characterized by challenging exploration but a simple policy.
>
>
> Analysis of reinforcement learning hardness is commonly carried out with linear models [3, 4, 5] in the literature to avoid overfitting to the relatively few data points. In addition to this, we also chose linear models with log transformation of the hardness measures because the regret bounds are usually sublinear function of those measures.
>
>
> ```
> In Section 4.3.3 the authors used a refined linear model to conduct the analysis and concluded that some variables were significant in for a class of environments, for example Breakout. This looks confusing to me as the authors did not define what is really a class of environments. How was the "breakout class" defined? Can the authors give more examples? I am expecting to see not just trivial definition of gridworlds with larger sizes, but actually a connection between different environments. If we had a reasonable definition for a class of Atari games, we could potentially use it to characterize the hardness even just empirically.
> ```
> Due to the lack of space, we could only provide representation of different instance of the environment classes in the Appendix. The reviewer can find several examples in Appendix B.
>
> A breakout environment instance is defined based on some characteristric of the game such as the number of rows, the number of columns, and the size of the paddle. For example, breakout instance with larger paddle sizes are easier because the agent can more easily hit the ball.
>
> In the freeway instance, it is possible to decide whether the agent being hit by a car induces a negative penalty or if resets the position of the agent at the start. The two options induce different levels of hardness with the latter being clearly significantly more challenging that former.
>
> If we understand your suggestion correctly, then yes, it would be possible to empirically measure how much harder each of those option make the environment harder for DQN (or any reinforcement learning agent). However, contrary to environment hardness measures (such as the diameter), this empirical measure would be specific to the DQN agent, and we have no theoretical explanation of how this would transfer to other agents.
>
>
> ```
> My another concern is that the paper revealed many issues of the existing protocols but did not propose a viable solution.
> This makes the paper seems rather incomplete to me: the authors showed correlation but did not suggest an usable metric. IMO, the paper could benefit from comprehensive evaluation on not just linear but also higher order models on a selected subset of variables to help conclude what might be an interesting combination as metric.
> ```
> The theoretical framework that would allow to provide a theoretically founded protocol are not yet available in the non-tabular setting.
> This paper represent a first step toward principle benchmarking of non-tabular reinforcement learning, but, as clearly identified by the reviewer, this is not as simple.
>
> The main contributions are the review of the theory of non-tabular hardness, the library (pharos), and the example of analysis that can be carried out with the library.
> Researchers building on top of our work will be better equipped to effectively study hardness in non-tabular reinforcement learning and potentially develop theoretically founded hardness measures that will be used to define a theoretically founded protocol.
>
> Please note that this paper is the first to try and systematically frame the problem of principled benchmarking in the non-tabular setting.

---

> > ### Author Response · Authors · 2024-12-15
> > **Response (2/2)**
> >
> > [1] Sutton, Richard S. "Reinforcement learning: An introduction." A Bradford Book (2018).
> >
> > [2] Mnih, Volodymyr. "Playing atari with deep reinforcement learning." arXiv preprint arXiv:1312.5602 (2013).
> >
> > [3] Conserva, Michelangelo, and Paulo Rauber. "Hardness in markov decision processes: Theory and practice." Advances in Neural Information Processing Systems 35 (2022): 14824-14838.
> >
> > [4] Aitchison, Matthew, Penny Sweetser, and Marcus Hutter. "Atari-5: Distilling the arcade learning environment down to five games." International Conference on Machine Learning. PMLR, 2023.
> >
> > [5] Laidlaw, Cassidy, Stuart J. Russell, and Anca Dragan. "Bridging rl theory and practice with the effective horizon." Advances in Neural Information Processing Systems 36 (2023): 58953-59007.

---

### Review · Reviewer_H8rU · 2024-11-29

**Summary Of Contributions:**

- This paper provides a compact survey of hardness measures for tabular RL problems and RL benchmarks.
- It identifies the desiderata for such measures for non-tabular RL, proposes a benchmarking tool, and conducts an empirical analysis of how well the proposed tool explains the hardrness of non-tabular RL environments.

**Audience:**

Yes

**Broader Impact Concerns:**

While RL as a whole may have broader impacts, there is nothing specific about the paper that would necessitate a deeper discussion of its impacts.

**Claims And Evidence:**

Yes

**Requested Changes:**

- It would be good to explain what the individual points in the plot represent in the caption for figure 4.
- Please explain more carefully the limitations of the proposed benchmarking tool. For example, it is not discussed how the tool would be applied for benchmarking existing algorithms.

### Typos etc
- A reference to the object being discussed is missing from the sentence wrapping equation 1.
- "The eluder dimension is generally defined as the longest possible sequence of tuple $(x_t, y_t)$" should have "tuples"
- "Access to state-of-the-art reinforcement learning agents is of paramount important" should have "of paramount importance".
- "For each environment instance, the DQN agent is trained with a simple mage-based representation and a simple vector-based representation, and the results are averaged across five seeds" while "mage-based" representations certainly sound fun, I'm guessing this should be image-based
- In figure 3, one panel is titled "brekout". Should probably be "breakout".

**Strengths And Weaknesses:**

- The analysis is thoughtful and supported by both theoretical and empirical evidence.
- The paper is mostly clearly presented. Though it would benefit from another round of proof reading (see requested changes for a non-exhaustive list of typos)
- The paper is a bit unclear about the framing of the proposed new benchmarking library. The new library is used for conducting the analysis in the paper, but it is not used for actual benchmarking for existing RL algorithms. The contributions are already valuable as is, but it would be good to mention this limitation.

---

> ### Author Response · Authors · 2024-12-15
>
> Thanks for reading the paper, and for the helpful comments.
>
> ```
> It would be good to explain what the individual points in the plot represent in the caption for figure 4.
> ```
> Thanks for raising the issue of lack of clarity on this.
>
> Each point in figure 4 and the following represents an environment instance from the specific environment classes, the reviewers can find visualizations of how those look like in Appendix B.
> We will add a sentence that refers to the Appendix B in the experimental setup paragraph.
>
> ```
> Please explain more carefully the limitations of the proposed benchmarking tool. For example, it is not discussed how the tool would be applied for benchmarking existing algorithms.
> ```
> The proposed tools (pharos) is a step toward a principled benchmarking tools but we can not yet claim that it can perform principled benchmarking because the theoretical framework that would allow to provide a theoretically founded protocol are not yet available in the non-tabular setting.

---

### Review · Reviewer_ViR2 · 2024-12-26

**Summary Of Contributions:**

This paper aims to explore benchmarking in non-tabular reinforcement learning. It reviews existing theories of environment hardness, identifies key features for designing a principled benchmark, proposes the Pharos library as a tool for such purposes, and conducts a case study to demonstrate its utility.

**Audience:**

Yes

**Broader Impact Concerns:**

No concerns about broader impacts.

**Claims And Evidence:**

No

**Requested Changes:**

1. Conduct a more comprehensive review of current benchmarks, analyzing their limitations in depth and connecting these shortcomings to the goals of this paper.
2. Include experiments with more environments and algorithms to better showcase the generalizability and practical impact of the proposed benchmarks.
3. Explicitly discuss the limitations of the proposed benchmarks and library. Typically, a good benchmark should be diverse, scalable, and representative of the real-world problems, with a set of environments that cover a wide range of difficulties yet share common characteristics. The current benchmarks proposed in the paper do not fully meet these criteria.

**Minor**
1. Section 2.2 lacks definitions for $H$ and $T$.
2. Section 4 (P8): Sentence "we analyse the results of training DQN agents with varying representation to highlight." is incomplete.
3. The last sentence in Sec. 1 lacks a period.

**Strengths And Weaknesses:**

**Strengths**

1. The paper summarizes existing theories of environment hardness, which is important for benchmarking non-tabular RL.
2. A new benchmarking library, Pharos, is introduced, to facilitate standardized benchmarking in non-tabular RL.

**Weaknesses**

1. The discussion of existing benchmarks is insufficient and lacks breadth.
2. The paper positions itself as a first step towards principled benchmarking, but its scope and limited empirical evidence do not fully justify this claim.
3. The benchmarks proposed lack diversity and do not align well with the scale and complexity of environments that RL researchers typically work with in recent years.

---

> ### Author Response · Authors · 2025-01-18
> **Response (1/1)**
>
> Thanks for reading the paper, and for the helpful comments!
> ```
> Conduct a more comprehensive review of current benchmarks, analyzing their limitations in depth and connecting these shortcomings to the goals of this paper.
> ```
> As stated in the title of the paper, we are not proposing a benchmark. The proposed tool (pharos) is a step toward a principled benchmarking tools but we can not yet claim that it can perform principled benchmarking because the theoretical framework that would allow to provide a theoretically founded protocol is not yet available in the non-tabular setting.
>
> However, given that there are related tools already available to practitioners, we do have an entire section (Sec. 3) dedicated to the review of those existing tools and an in-depth comparison.
>
> ```
> Include experiments with more environments and algorithms to better showcase the generalizability and practical impact of the proposed benchmarks.
> ```
> The key reason why we focused on a relatively small selection of environments and a single agent is that we are prioritizing a detailed qualitative analysis of the experiments rather than a large scale quantitative one, which could be a very interesting future work.
>
> This fits well with the goal of the paper, which is a step towards principled benchmarking rather than a new benchmark.
>
> However, to make sure that the results are not too specific to DQN, we have run a PPO with a similar experimental setup and we have found that the performances of the two agents are highly correlated (0.81) in the pharos environments.
>
> ```
> Explicitly discuss the limitations of the proposed benchmarks and library. Typically, a good benchmark should be diverse, scalable, and representative of the real-world problems, with a set of environments that cover a wide range of difficulties yet share common characteristics. The current benchmarks proposed in the paper do not fully meet these criteria.
> ```
> Thanks for raising this point. We have added limitations section for the tools and for the experiments. We have added them here for clarity. Please note that, given the focus of the paper, the limitations are focused on the use of the tools to analyze hardness rather than as a benchmarking library.
>
> ### **Limitation paragraph from section discussing the tool (Sec. 3)**
> A current limitation of pharos is the relatively small number of available environment classes.
> While the existing classes offer some flexibility in customization, this limitation may restrict the diversity of the hardness analysis.
> While pharos aims to simplify the process of creating additional environment, this still presents an obstacle to leveraging environments already available in other libraries.
> Future development should prioritize expanding the variety of built-in environment classes and consider building bridges with other environment libraries to facilitate broader exploration and evaluation.
>
> ### **Limitation paragraph from section discussing the results (Sec. 4)**
> This study focused on a limited set of environments and a single agent to allow for an in-depth qualitative analysis of the results.
> Future work could expand the scope of this research by including a wider range of agents, environments, and representation types, which would further validate the findings and enhance the generalizability of the results.
> To partially address this limitation, we have run the same setup for a PPO agent and found that the performances of the two agents are highly correlated (`0.81`).
> Tables describing the full results for the two agents are available in Appendix C.

---

### Decision · Action_Editor_3nCa · 2025-03-04

**Recommendation:** Reject

**Comment:**

While the paper is addressing a very relevant problem it does not provide sufficiently convincing evidence and clarity to be accepted in the current form - see my comments regarding the claims above. I did not consider the breadth of the empirical evaluation too limited regarding TMLR's acceptance criteria. If the authors want to submit a revised version, they should improve clarity in line with the reviewers comments and, in particular, more clearly show how the proposed library can support the design of "principled benchmarking" for non-tabular RL.

Other comments:
The authors should consider adjusting the title to be in line with the expectations of readers.

**Audience:**

Yes, the paper is relevant to individuals evaluating reinforcement learning algorithms and in particular to those working on building benchmarks for such evaluation.

**Claims And Evidence:**

The submitted paper makes four explicit claim and an implicit claim through its title.
1.) "Review the theory of hardness in tabular and non-tabular settings to highlight promising directions": This review is performed in section 2 and appreciated by the reviewers.
2) "identify the essential features that a principled benchmarking library for non-tabular reinforcement learning should possess while explaining the limitations of existing libraries in meeting those needs": This aspect is discussed in section 3 but for instance reviewer ViR2 considers the the discussion of existing benchmarks to be "insufficient and lacks breadth". This was not addressed by the authors in the rebuttal.
3) "propose a new library (pharos) specifically designed to support the development of principled benchmarking": This claim is challenging to assess as it is somewhat unclear to me how to judge that the library supports the development of the principled benchmarking. My unclarity in this regard seems to be shared by the reviewers and is amplified by statements of the authors like the following: "The proposed tools (pharos) is a step toward a principled benchmarking tools but we can not yet claim that it can perform principled benchmarking because the theoretical framework that would allow to provide a theoretically founded protocol are not yet available in the non-tabular setting."
4) "present an in-depth case study": While the case study is interesting, I found the relation with pharos quite confusing, as also hinted at by reviewer H8rU. Upon careful reading what is claimed as the contribution, this, however, does not contradict the contribution which does not mention the usage of pharos.
5) Title: Two reviewers considered the paper to not live up to the "towards principled benchmarking" promised in the title. While it is hard to clearly specify what to expect from a paper with such a title, it seems expectations are not met.

**Resubmission Of Major Revision:**

The authors may consider submitting a major revision at a later time.